# Chasing the Role of miRNAs in RCC: From Free-Circulating to Extracellular-Vesicle-Derived Biomarkers

**DOI:** 10.3390/biology12060877

**Published:** 2023-06-17

**Authors:** Ilenia Mastrolia, Virginia Catani, Marco Oltrecolli, Stefania Pipitone, Maria Giuseppa Vitale, Valentina Masciale, Chiara Chiavelli, Carlo Augusto Bortolotti, Cecilia Nasso, Giulia Grisendi, Roberto Sabbatini, Massimo Dominici

**Affiliations:** 1Laboratory of Cellular Therapy, Division of Oncology, Department of Medical and Surgical Sciences for Children & Adults, University of Modena and Reggio Emilia, 41124 Modena, Italy; virginia.catani@unimore.it (V.C.); valentina.masciale@unimore.it (V.M.); chiara.chiavelli@unimore.it (C.C.); giulia.grisendi@unimore.it (G.G.); 2Division of Oncology, Department of Oncology and Hematology, University Hospital of Modena, 41124 Modena, Italy; oltrecolli.marco@gmail.com (M.O.); stefania.pipitone88@gmail.com (S.P.); vitale.mariagiuseppa@aou.mo.it (M.G.V.); cecilianasso30@gmail.com (C.N.); sabbrob@unimore.it (R.S.); 3Department of Life Sciences, University of Modena and Reggio Emilia, 41124 Modena, Italy; carloaugusto.bortolotti@unimore.it; 4Division of Oncology, S. Corona Hospital, 17027 Pietra Ligure, Italy

**Keywords:** renal cell carcinoma, liquid biopsy, blood, extracellular vesicles, microRNA, biomarker, diagnostics, prognosis, prediction

## Abstract

**Simple Summary:**

Renal cell carcinoma (RCC) is the most common type of kidney cancer. The therapeutic strategies are based on surgery and/or specific therapies able to inhibit growth factors that have been shown to promote the growth and spread of tumors. Currently, there is no established biomarker which helps in early diagnosis and in better disease monitoring with a high sensitivity. Much information could be provided by body fluids, especially blood liquid biopsy (LB), that are increasingly interesting to researchers. LB is a non- or minimally invasive procedure that could allow clinicians to monitor cancer evolution, also thanks to the presence of small vesicles known as extracellular vesicles (EVs) secreted by tumor cells and containing useful information. In particular, growing interest is focused on small RNA molecules (miRNAs) that are involved in tumor growth and could represent potential diagnostic, prognostic, and predictive biomarkers in RCC, as we summarize in this review.

**Abstract:**

Renal cell carcinoma (RCC) is the second most common cancer of the urinary system. The current therapeutic strategies are based on partial or total nephrectomy and/or targeted therapies based on immune checkpoint inhibitors to which patients are often refractory. Preventive and screening strategies do not exist and the few available biomarkers for RCC are characterized by a lack of sensitivity, outlining the need for novel noninvasive and sensitive biomarkers for early diagnosis and better disease monitoring. Blood liquid biopsy (LB) is a non- or minimally invasive procedure for a more representative view of tumor heterogeneity than a tissue biopsy, potentially allowing the real-time monitoring of cancer evolution. Growing interest is focused on the extracellular vesicles (EVs) secreted by either healthy or tumoral cells and recovered in a variety of biological matrices, blood included. EVs are involved in cell-to-cell crosstalk transferring their mRNAs, microRNAs (miRNAs), and protein content. In particular, transferred miRNAs may regulate tumorigenesis and proliferation also impacting resistance to apoptosis, thus representing potential useful biomarkers. Here, we present the latest efforts in the identification of circulating miRNAs in blood samples, focusing on the potential use of EV-derived miRNAs as RCC diagnostic and prognostic markers.

## 1. Renal Cell Carcinoma: Clinical Diagnosis and Staging

Renal cell carcinoma (RCC) is the second most lethal malignancy in the urological system [1] and represents about 3% of all tumors affecting adults in the Western world [2]. Every year, 64,000 new RCC cases are diagnosed, and due to this malignancy, 14,000 mortality cases are registered in the United States (US), with it projected to increase in burden worldwide [3]. Early RCC diagnosis is critical to reduce mortality rate; however, the disease is characterized by an asymptomatic clinical course, with most cases accidentally diagnosed during clinical evaluations for other concomitant pathologies [2]. Nevertheless, some clinical signs could be related to RCC, such as polycythemia, hypercalcemia, hypertension, and Cushing’s syndrome, correlated to erythropoietin, parathyroid-related hormone, renin, and adrenocorticotropic hormone (ACTH) overproduction by cancer, respectively. Furthermore, leukocytosis, fever, and weight loss are common symptoms, while only about 10% of cases show the “classic triad” of RCC symptomatology, consisting of flank pain, hematuria, and palpable masses [4].

The incidence of the disease varies widely throughout the world and appears to be highest in more developed countries. White people have a lower risk of RCC than Native, African, and Hispanic Americans, probably due to the disparities in diets, lifestyles, socioeconomic conditions, education, and economical possibility for healthcare. Risk factors connected to RCC could be distinguished as (a) unmodifiable, such as age, race, and gender, and (b) modifiable, that include diet, alcohol, occupational exposure, and drugs [4]. The peak incidence is mostly around 60 and 80 years old, when the average is 64 years in the US [3]. 

Unmodifiable and modifiable risk factors are often related: RCC is doubly diagnosed in men where some modifiable risks such as hypertension, tobacco smoking, and obesity are more frequently observed. 

RCC originates from the kidney cortex, in particular from the epithelial cells of renal tubules [3,4], and is commonly classified into three main histopathological subtypes: (1) Clear cell renal cell carcinoma (ccRCC) represents 75% of all RCC cases and 90% of all RCC deaths due to its tendency to metastasize [2]; it derives from renal cancer progenitors, namely RCC cancer stem cells (CSCs), and it is often associated with distant disease to the bones, liver, and lungs through hematologic ways [4]. (2) Around 10% of diagnosis consists of papillary renal cell carcinoma (pRCC), characterized by histological subclassification in type I and II, where the first has a tropism for lymphatic recurrence. (3) Chromophobe renal cell carcinoma (chRCC) is the less aggressive subtype and represents 5% of all RCC [4]. The RCC stage at diagnosis is an important prognostic factor. Indeed, for organ-confined tumors (stage I), over 90% of individuals remain surviving at 5 years from onset, while the survival percentage is reduced to 72.5% for regional disease (stage II/III). Unfortunately, about 30% of RCC cases are diagnosed at metastatic disease stage (stage IV), where the prognosis is poor with a 5-year survival rate of 12% [4].

## 2. RCC Treatments: Advances and Challenges in the Lack of Biomarkers

RCC is a chemo- and radio-resistant neoplasia. The current therapeutic strategies are based on the surgical approach of radical and/or partial nephrectomy that remains the only effective therapy for clinically localized RCC [5]. Traditionally, these tumors have been treated aggressively, most often with radical nephrectomy; however, it predisposes patients to chronic kidney disease with associated cardiovascular risks and increased mortality [6]. Sparing approaches such as partial nephrectomy, thermal ablation, and active surveillance have also emerged as viable options for the management of these patients [7]. Treatment options are, thus, individualized and related to disease features (above all, tumor stage, size, and grade) [2,4,8], weighing the risks and benefits of partial versus radical nephrectomy. Nevertheless, RCC is characterized by a high rate of relapse and, unfortunately, about 20–50% of those treated with curative intent will progress to stage IV. The features related to a worse prognosis are the tumor size (>7 cm), tumor necrosis, a fat invasion, and the performance status (Karnofsky Performance Scale Index (KPS) < 80) together with nodal metastases and age [4]. The lack of sensitivity to chemotherapy and radiotherapy has driven research efforts towards novel therapeutic options. In the last few years, the therapeutic approach for metastatic RCC (mRCC) in the first-line setting has radically changed due to novel combinatory approaches impacting both immunity and intracellular molecular pathway inhibition.

Those approaches target the peculiar hypervascularity of RCC and trigger immune cell infiltration by immune checkpoint inhibitors (ICIs), with or without antiangiogenic tyrosine kinase inhibitor (TKI)-based combinations [9]. Thus, the current standard of care for frontline advanced ccRCC includes an “immuno-oncology” (IO) combination that may include a vascular endothelial growth factor (VEGF) inhibitor with a checkpoint inhibitor (VEGF-IO), or two checkpoint inhibitors together (IO-IO), which showed efficacy and safety [10,11]. Interestingly, antiangiogenic TKIs in RCC are given without actionable mutations to screen. They are oral drugs able to block the angiogenesis pathway by binding the VEGF receptor, mesenchymal–epithelial transition factor (c-MET), and rearranged during transfection (RET) tyrosine kinase receptor. For RCC, TKIs can be classified into three generations of drugs, according to the spectrum of action and side effects: sunitinib, pazopanib, and sorafenib (first generation); axitinib and tivozanib (second generation); and cabozantinib (third generation) [12]. 

ICIs can stimulate the host immune response against cancer through PD1 inhibitors (Nivolumab and Pembrolizumab) or anticytotoxic T-lymphocyte-associated protein 4 (CTLA4) inhibitors (Ipilimumab). The first group can block the interaction between programmed cell death protein 1 (PD1) and programmed death-ligand 1 (PD-L1) able to downregulate the host T cells and, therefore, the immune response [11]. Many studies have investigated the role of PD-L1 expression as a biomarker, such as Iacovelli et al. [13], who performed a meta-analysis of six studies on 1323 patients, showing a high level of PD-L1 expression associated with a higher risk of death. Indeed, PD-L1 expression may be considered a negative prognostic factor in RCC [14].

However, the response and tolerance to these combination treatments are not always satisfying, probably due to the lack of recognized molecular targets, an intratumoral heterogeneity, and different RCC molecular subtypes. 

The therapeutic strategy is based on the clinical prognostic score (International Metastatic RCC Database Consortium (IMDC) and Memorial Sloan Kettering Cancer Center (MSKK) criteria), drug toxicities, patients’ comorbidities, and preference. In fact, nowadays the International mRCC Database Consortium risk model remains the only prospectively known predictive biomarker in mRCC [15].

Although targeted therapies represent the standard treatment options for RCC, nearly all patients treated with targeted drugs will eventually experience disease progression. Unfortunately, a significant number of RCC patients are primarily refractory to targeted therapeutics, showing neither disease stabilization nor clinical benefits, probably due to the intratumoral heterogeneity and molecular subtypes [15,16].

Therefore, the identification of new promising biomarkers in RCC is an urgent clinical need that leads clinicians and researchers to investigate the role of cell components as microRNAs (miRNAs) secreted by tumor cells and packaged into extracellular vesicles (EVs) released in biological fluids [17].

## 3. Liquid Biopsy in RCC

For decades, surgical tissue biopsy has firmly been considered the gold standard for solid tumor diagnosis; however, it may become a diagnostic obstacle for its invasiveness, unrepeatability, and especially its peculiarity to immortalize that moment without taking into account the tumor heterogeneity and dynamism [18]. The molecular tumor pattern may dynamically evolve over time, driven by microenvironmental stimuli and treatment pressure, rendering future tissue-biopsy-based therapeutic decisions suboptimal [19,20].

Over the past decade, liquid biopsy (LB) has revolutionized the field of clinical oncology, providing ease in tumor sampling and continuous tumor monitoring [21].

Thus, LB is gaining relevance in the clinical scenario of cancer for early diagnosis and treatment stratification, as well as residual disease detection and recurrence monitoring [20]. 

In particular, overall survival in mRCC patients remains low despite the development of novel targeted therapeutic approaches. LBs could provide an attractive and noninvasive method to determine the risk of recurrence or metastatic dissemination during follow-up and, thus, improve the clinical outcomes and quality of life of RCC patients [22]. Several biological body fluids, such as blood, urine, saliva, breast milk, and other patient samples, can be collected and used for clinical investigations as they contain numerous biomarkers. Peripheral blood, as well as other biological fluids, can be a source of cancer material, such as circulating tumor cells (CTCs), circulating tumor DNA (ctDNA), and tumor-derived EVs [23] (Figure 1).

The identification of biomarkers in LB could enable the continuous and real-time collection of RCC patients’ information for diagnosis, prognosis assessment, and treatment response monitoring [24].

Many studies have attempted to use LB as a routine method for the clinical diagnosis of RCC and for the prediction of patients’ grades, stages, and survival to identify patients at high risk for metastasis and recurrence [25,26,27,28,29]. In particular, in recent decades, most efforts have been spent in analyzing CTCs and ctDNA [30,31]; however, their low blood levels make them very difficult to catch and analyze [24]. 

EVs and miRNAs are emerging as a platform with potentially broader and complementary applications in LB [32]. 

## 4. Extracellular Vesicles: Biogenesis and Role in RCC

EVs are small vesicles enclosed in a lipid bilayer, continuously released by all viable cells, in physiological and pathological conditions [33]. According to the international society for extracellular vesicles (ISEV) guidelines, the term EVs includes three different subtypes of vesicles based on their biogenesis and size [33]: (a) exosomes have a diameter of 30–150 nm and originate from the fusion between intracytoplasmic multivesicular bodies (MVB) and the cellular plasma membrane; (b) microvesicles with a size of 100–1000 nm are generated directly from the plasma membrane through, the budding process; (c) apoptotic bodies are larger than 1000 nm and are released by dying cells [34] (Figure 2). A list of minimal information for studies of EVs was provided from minimal information for studies of extracellular (MISEV)-2014 [35] and updated in MISEV-2018 [34] pointing out EV separation/isolation and characterization techniques and functional studies. 

EVs can transport tumor-derived material, including lipids, a wide variety of RNA (miRNAs, mRNAs, and long noncoding RNAs), DNA, proteins, enzymes, and metabolites. They play a key role in cell-to-cell communication [36]; by transferring their information to nearby or long-distance recipient cells, influencing their phenotype and functions [37]. 

These small messengers are involved in tumorigenesis and tumor metastasis due to their widespread distribution throughout the blood and lymphatic circulation. Several studies indicate that tumor-released EVs may orchestrate tumor-progression-stimulating proliferation, angiogenesis, chemoresistance, and immune escape [19,38,39]. In addition, EVs are highly stable in biological fluids and protect their content from RNase and protease activity, thanks to their lipid bilayer structure [40].

Recently, EVs showed increasing potential in the field of urological malignancies [41]. An increasing number of studies are focusing on EV cargoes, particularly miRNAs, in relation to diagnostic and prognostic accuracy, treatment response, as well as numerous biological processes [42,43]. 

The recent knowledge is related to the biological role of EVs shed by RCC tumor cells in RCC progression, such as angiogenesis, immune escape, and tumor growth [17].

Horie K et al. analyzed the hypoxic conditions in RCC that stimulate the release of tumor-cell-derived EVs. Their in vitro results, based on migration and tube formation assays, suggested the possibility that carbonic anhydrase 9 (CA9) enriched in exosomes and released from hypoxic RCC may enhance angiogenesis in the microenvironment, thereby contributing to cancer progression [44]. In addition, RCC-EVs were studied for their modulation of vascular permeability affecting metastatization, thanks to their enrichment in the azurocidin protein (AZU1) being significantly higher in serum and cancer-tissue-derived EVs from ccRCC patients compared to those from healthy donors. They tested the in vitro permeabilization ability of EVs derived from cancer tissues and adjacent normal tissues, showing that tumor EV upregulated cell permeability [45].

Moreover, specific protumorigenic roles have been attributed to EVs released by CD105+ renal CSCs, due to their cargo enriched in several proangiogenic mRNAs and miRNAs [38]. Those cargoes can also impact other tumor microenvironmental cells, as demonstrated by Lindoso et al., considering the involvement of renal-CSC-derived EVs in the interaction between tumor and stroma. They found that CSC-derived EVs promoted phenotypical changes in stromal cells through an increased expression of genes associated with cell migration, matrix remodeling, angiogenesis, and tumor growth [39]. Grange et al. additionally demonstrated the impact of CSC-derived EVs isolated from RCC as the main mediators for the additional tumor microenvironment element, impacting monocyte differentiation into dendritic cells [46].

A specific role of renal-CSC-derived EVs has been additionally described for the formation of the premetastatic niche, which consists of a complex network of information exchanges. These EVs may sustain an unfavorable outcome for the tumor by enhancing tumor vascularization and by contributing to the establishment of a premetastatic niche [38]. More recently, it has been demonstrated that renal CSC-EVs were enriched by miRNAs influencing cell growth, tumor invasion, and metastases, thus giving to circulating miRNAs a promising role as biomarkers in RCC [17].

## 5. MicroRNAs (miRNAs): From Free-Circulating to EV-Packaged Biomarkers

MicroRNAs (miRNAs) are small noncoding RNA molecules of 18–22 nucleotides capable of post-transcriptionally regulating protein translation, due to increased mRNA degradation [37,38]. For this reason, the biological role of miRNAs is usually associated with the role of their target mRNAs. Since a single miRNA can positively or negatively modulate numerous target mRNAs, they emerge as biological regulators of many cell functions including cell growth and proliferation and resistance to apoptosis; therefore, they play a key role in tumorigenesis and tumor progression, representing potentially useful biomarkers [2,47,48]. Because abnormal cell proliferation is a hallmark of all cancers, the miRNA expression patterns might denote the malignant state.

Over the last few years, the aberrant expression of miRNAs in cancer development has been dissected, identifying some miRNAs highly expressed in cancer cells with an oncogenic role and miRNAs acting as tumor suppressors. The deregulation of both groups of miRNAs contributes to tumor development mechanisms: oncogenic miRNAs (onco-miRNAs) are commonly overexpressed in cancerous tissues, and tumor suppressor miRNAs are conversely downregulated in tumors [2]. Their stability in body fluids increases when they are packaged within EVs [49]. The utility of circulating miRNAs in RCC has been previously evaluated [50]; nevertheless, recently, further evidence has been added to this emerging field, focusing on EV-encapsulated miRNAs from blood samples.

In 2017, for the first time, Tian and colleagues compared the expression levels of miRNAs in plasma and in plasma-derived exosomes, showing no differences in healthy donors. However, in lung cancer patients, the onco-miRNAs were more enriched in exosomes than free in circulation, assuming an increased involvement of EVs in cancer for miRNA-based cell-to-cell communication [51].

Cancer cells can actively load miRNAs into EVs for supporting tumor development and spread due to their peculiarity of protecting their cargo from the environment. The EV double layer allows the content to be preserved, thus improving miRNA’s half-life and stability [52]. Furthermore, EVs secreted by tumor cells may be distinguished from the others by the presence of specific tumor markers on their surface, becoming a source of precious information in tumor diseases [53].

In RCC, many efforts have been made in the research of circulating miRNAs in biological fluids, such as serum, plasma, and urine. Because urine can be easily collected, urinary miRNAs are attractive as they may be promising biomarkers in diseases affecting the urinary tract. Several studies demonstrated their stability in urine, particularly when carried by EVs. However, urine samples and, consequently, urinary miRNA detection may be influenced by several physiological and pathological conditions which make their study very challenging [54,55].

Due to the ease of sampling, better specificity of detection, and higher stability, most studies focused on serum and plasma miRNAs as potential noninvasive diagnostic, prognostic, or predictive biomarkers in RCC patients [53].

## 6. The Potential Role of Blood-Circulating miRNAs in RCC

The expression profile of several miRNAs has revealed their valid potential as diagnostic biomarkers in human RCC, since they are differentially expressed in tumors when compared with their normal counterpart, but also between primary and metastatic tumors [2]. They have great clinical relevance in RCC since a considerable number of ccRCC patients are diagnosed with metastatic disease, reducing the effectiveness of actual therapies. Indeed, a late diagnosis can limit treatment options, often leading to a poor prognosis. For this reason, the search for novel molecular biomarkers can significantly improve early RCC diagnosis and prognosis. Experimental evidence has also revealed the potential prognostic value of different miRNAs correlated with the overall survival of RCC patients. However, the miRNA prognostic signature is debated due to the lack of uniformity among studies regarding the identification of clinically significant miRNAs. The discrepancy may be due to the use of different cohorts of patients, so comorbid conditions and genetic differences, which are unrelated to RCC, may substantially influence the results [56].

An increasing challenge in recent studies is to find potential biomarkers able to improve sensitivity to chemoradiotherapy, predicting patient’s response and resistance to targeted therapy. Circulating miRNAs might also be involved in acquiring resistance to treatment in RCC. Their promising role as predictors of response to therapy may acquire considerable importance in RCC, providing clinicians with crucial information to determine the optimal treatment plan and, thus, avoiding severe side effects of ineffective overtreatment [57] (Figure 3).

We reviewed the studies on free or encapsulated miRNAs differentially expressed in the blood samples of RCC patients compared to normal controls, as summarized in Table 1. We observed that most of the studies analyzed the miRNAs’ expression within serum and plasma, without focusing on EV-derived miRNAs.

Tusong et al. reported the overexpression of miR-21 and miR-106a in RCC patients’ serum compared to healthy controls and their significant reduction a month after surgery compared with the preoperative group. This outcome suggests the potential diagnostic and predictive role of serum miR-21 and miR-106a as noninvasive biomarkers for RCC [58]. MiR-21 has been given much attention, due to an important role in a variety of malignant tumors. The correlation between the upregulation of miR-21 in the serum of ccRCC patients and their clinical stage was described in the literature [59]; however, the underlying mechanisms are not well understood. Liu et al. reported that upregulated miR-21 in serum predicates advanced clinic–pathological features and poor prognosis in patients with RCC through the p53/p21-cyclin E2-Bax/caspase-3 signaling pathway [60].

Cheng et al. showed an upregulation of miR-21, miR-34a, and miR-224 and a downregulation of miR-141 in the sera of patients with ccRCC, reporting consistency in their expression profile with those of the corresponding tumor tissue samples [59].

Interestingly, only the miR-21 expression levels in the serum of the patients were correlated with the patients’ clinical stage, while miR-224 expression levels were correlated with gender. However, this study showed that miR-34a, miR-224, miR-21, and miR-141 could be considered as potential ccRCC tumor markers [59].

A diagnostic role has been attributed to several miRNAs, such as: miR-625-3p, miR-508-3p, and miR-885-5p, dysregulated in the serum of ccRCC patients [61,62]; miR-218 and miR-222, upregulated in the plasma of patients [63,64]; miR-149-3p, miR-424-3p, and miR-92a-1-5p, which are significantly abnormal in exosomes from the plasma of RCC patients [65]; and miR-1233, which appeared upregulated in the serum [66], exosomes from serum [67], and plasma [63] samples of RCC patients. Huang et al. identified miR-182-5p, miR-224-5p, and miR-34b-3p as potential diagnostic biomarkers after three-stage selection [68]; similarly, a diagnostic role was successfully validated by Redova et al. for miR-378 and miR-451 due to the increased and decreased expression, respectively, in the serum of RCC patients, enabling their potential use as biomarkers for distinguishing between RCC and healthy controls [69].

The late diagnosis of RCC, primarily due to a lack of early-stage diagnosis, encouraged researchers to discover novel miRNA signatures that discriminate RCC patients from healthy controls with a high degree of accuracy. In particular, in 2015, Whang et al. focused on a 5-miRNA panel including miR-28-5p, miR-362, miR-572, miR-193a-3p, and miR-378, demonstrating their potential value as an ancillary clinical diagnostic tool to also detect early-stage RCC, for which surgery is most effective [70]. Next to a diagnostic role, in the same year, other researchers dissected the prognostic and predictive value of miR-378 together with miR-210, describing their significant decrease in the time period of three months after radical nephrectomy [63,67,71,72,73].

More recently, the upregulation of miR-765 was described in the plasma of ccRCC patients after tumor resection, suggesting its tumor suppressor role and identifying the proteolipid protein 2 (PLP2) as a candidate downstream target gene [74]. In the same year, Huang and his team suggested a diagnostic ability in miR-196a, miR-20b-5p, and miR-30a-5p in the serum of 110 RCC patients and 110 healthy controls. Additionally, analyzing the clinical role of each miRNA, they showed a significant correlation between miR-196a-5p expression and the Fuhrman grade and clinical stage, demonstrating its potential involvement in the oncogenesis and tumor progression of RCC [75]. Diagnostic and prognostic significance was defined for miR-144-3p [76], miR-187 [77], miR-221 [63,64], and miR-183-5p. In particular, it was demonstrated that the cells of patients with high levels of this miRNA in the blood poorly responded to the cytotoxicity induced by natural killer (NK) cells, suggesting that targeting miR-183-5p may be an effective way to enhance the outcome of NK-cell-based immunotherapy [78].

In contrast to other studies, Heinemann et al. did not only compare the serum of patients with ccRCC and healthy donors, but also a group of patients with benign renal tumors (BRTs). As a result, the downregulation of miR-122-5p and miR-206 in the serum of both ccRCC and BRT groups compared to controls suggested their doubtful role in diagnosis. However, they demonstrated their prognostic value, showing that high levels of miR-122-5p and miR-206 were associated with a significantly shorter period of progression-free, cancer-specific, and overall survival [79]. Plasma miRNA signatures specifically associated with late-stage disease were provided by researchers, contributing evidence that circulating miRNAs, such as miR-150 and miR-let-7b-5p, are associated with the progression of renal carcinoma and may contribute to disease monitoring [80,81].

The comparison of plasma of ccRCC patients with localized and metastatic disease before and after surgery suggested that EV content varies depending on the presence or absence of the disease. An increased level of miR-301a-3p and a decreased expression of miR-1293 may be potential biomarkers of metastatic disease [82]. Two studies, encompassing two distinct miRNAs (specifically miR-22 and miR-483-5p), significantly related their deregulation with the tumor stage and clinicopathological parameters, showing diagnostic, prognostic, and predictive functions [83,84]. Taken together, these findings revealed the potential role of miRNAs as promising biomarkers in RCC. They could provide a novel therapeutic point of view and their translation into clinical practice would be of great relevance in this malignancy.

**Table 1 biology-12-00877-t001:** Deregulated miRNAs in blood samples of RCC patients as potential diagnostic, prognostic, or predictive biomarkers.

miRNA	Expression Changes in RCC	Source	Therapeutic Role	Reference
miR-106	Upregulated	Serum	Diagnostic and predictive	Tusong et al., 2017 [58]
miR-122-5p	Upregulated	Serum	Prognostic	Heinemann et al., 2018 [79]
miR-1233	Upregulated	Exosomes— serum	Diagnostic and predictive	Zhang et al., 2018 [67]
	Upregulated	Plasma	Diagnostic and prognostic	Dias et al., 2017 [63]
	Upregulated	Serum	Diagnostic	Wulfken et al., 2011 [66]
miR-141	Downregulated	Serum	Diagnostic	Cheng et al., 2013 [59]
miR-144-3p	Upregulated	Plasma	Diagnostic and prognostic	Lou et al., 2017 [76]
miR-149-3p	Upregulated	Exosomes— plasma	Diagnostic	Xiao et al., 2020 [65]
miR-150	Downregulated	Plasma	Prognostic	Chanudet et al., 2017 [80]
miR-182-5p	Downregulated	Serum	Diagnostic	Huang et al., 2020 [68]
miR-187	Downregulated	Plasma	Diagnostic and prognostic	Zhao et al., 2013 [77]
miR-193a-3p	Upregulated	Serum	Diagnostic	Wang et al., 2015 [70]
miR-196a	Upregulated	Serum	Diagnostic and prognostic	Huang et al., 2020 [75]
miR-206	Upregulated	Serum	Prognostic	Heinemann et al., 2018 [79]
miR-20b-5p	Downregulated	Serum	Diagnostic	Huang et al., 2020 [75]
miR-21	Upregulated	Serum	Diagnostic and predictive	Tusong et al., 2017 [58]
	Upregulated	Serum	Diagnostic and prognostic	Cheng et al., 2013 [59]
	Upregulated	Serum	Diagnostic and prognostic	Liu et al., 2017 [60]
miR-210	Upregulated	Serum	Diagnostic and predictive	Fedorko et al., 2015 [71]
	Upregulated	Exosomes— serum	Diagnostic and predictive	Zhang et al., 2018 671]
	Upregulated	Plasma	Diagnostic and prognostic	Dias et al., 2017 [63]
	Upregulated	Serum	Diagnostic	Iwamoto et al., 2014 [73]
	Upregulated	Serum	Diagnostic and predictive	Zhao et al., 2013 [72]
miR-218	Upregulated	Plasma	Diagnostic	Dias et al., 2017 [63]
miR-22	Downregulated	Serum	Diagnostic, prognostic, and predictive	Li et al., 2017 [84]
miR-221	Upregulated	Plasma	Diagnostic and prognostic	Teixeira et al., 2014 [64]
	Upregulated	Plasma	Diagnostic and prognostic	Dias et al., 2017 [63]
miR-222	Upregulated	Plasma	Diagnostic	Teixeira et al., 2014 [64]
miR-224	Upregulated	Serum	Diagnostic	Cheng et al., 2013 [59]
	Upregulated	Serum	Diagnostic	Huang et al., 2020 [68]
miR-28-5p	Downregulated	Serum	Diagnostic	Wang et al., 2015 [70]
miR-30a-5p	Downregulated	Serum	Diagnostic	Huang et al., 2020 [75]
miR-34a	Upregulated	Serum	Diagnostic	Cheng et al., 2013 [59]
miR-34b-3p	Downregulated	Serum	Diagnostic	Huang et al., 2020 [68]
miR-362	Upregulated	Serum	Diagnostic	Wang et al., 2015 [70]
miR-378	Upregulated	Serum	Diagnostic and predictive	Fedorko et al., 2015 [71]
	Downregulated	Serum	Diagnostic	Wang et al., 2015 [70]
	Upregulated	Serum	Diagnostic	Redova et al., 2012 [69]
miR-424-3p	Upregulated	Exosomes— plasma	Diagnostic	Xiao et al., 2020 [65]
miR-451	Downregulated	Serum	Diagnostic	Redova et al., 2012 [69]
miR-483-5p	Downregulated	Plasma	Diagnostic, prognostic, and predictive	Wang et al., 2021 [83]
miR-508-3p	Downregulated	Serum	Diagnostic	Liu et al., 2019 [61]
miR-572	Upregulated	Serum	Diagnostic	Wang et al., 2015 [70]
miR-625-3p	Downregulated	Serum	Diagnostic	Zhao et al., 2019 [62]
miR-765	Downregulated	Plasma	Predictive	Xiao et al., 2020 [74]
miR-885-5p	Upregulated	Serum	Diagnostic	Liu et al., 2019 [61]
miR-92a-1-5p	Downregulated	Exosomes— plasma	Diagnostic	Xiao et al., 2020 [65]
miR-1293	Downregulated	EVs— plasma	Prognostic and predictive	Dias et al., 2020 [82]
miR-301a-3p	Upregulated	EVs— plasma	Prognostic and predictive	Dias et al., 2020 [82]
miR-let-7i-5p	Downregulated	Exosomes— plasma	Prognostic	Du et al., 2017 [81]
miR-183-5p	Upregulated	Serum	Diagnostic and prognostic	Zhang et al., 2015 [78]

## 7. Conclusions

RCC is considered one of the most unfavorable tumor diseases due to the late diagnosis and the poor prognosis. Nowadays, specific biomarkers validated for RCC early detection are not available and the actual treatments are often unable to avoid recurrence of the disease. LB could provide an attractive and noninvasive tool to assist the research for biomarkers in RCC tumors to capture a larger amount of the molecular heterogeneity compared to tissue biopsy and give prompt information on the risk of recurrence/relapse during follow-up.

Many efforts are directed towards the research for circulating miRNAs that play a role in almost all aspects of cancer biology and development. In particular, miRNAs packaged in EVs and released in blood flow could allow expanding the spectrum of potential biomarkers for future use in the diagnosis and prognosis of RCC and in the prediction of therapeutic response.

However, nowadays, there are no miRNAs widely applied as biomarkers in the clinical setting, partly due to the lack of isolation and quantification standardized protocols, the heterogeneity of study cohorts, and the variety of body fluids under investigation. Several studies have analyzed plasma- and serum-derived miRNAs, without focusing on the vesicle compartment. EV-derived miRNAs appear to be more stable than free miRNAs as EVs seem to protect them from degradation by macrophages. The EVs’ double-layered membrane and nanoscale size grant the miRNAs stability, thus prolonging their circulation half-life and enhancing their biological activity [85]. After release, EVs are taken up by neighboring or distant cells, and the encapsulated miRNAs modulate several processes, such as interfering with tumor immunity and the microenvironment, possibly facilitating tumor growth, invasion, metastasis, angiogenesis, and drug resistance [86]. Therefore, EV-derived miRNAs have a significant function in regulating cancer progression. More investigations into this matter are warranted. Moreover, the major challenges in studying miRNAs in biological fluids are related to the inconsistencies of miRNA signatures, not only between different studies, but also between different biological fluids of the same patient. This can be, in part, related to the lack of standardized protocols, especially when miRNAs are investigated within circulating EVs. There are numerous EV isolation protocols, which may affect the yield and stability of the miRNAs therein. In addition, a major limitation is the selection of proper miRNA reference controls for normalization, which becomes fundamental when the quantity of miRNAs is not directly proportional to the starting volume of biological fluid. In the literature, a miRNA used as a normalizer in one pathology is instead reported as a target in another one. To date, the lack of a highly conserved and universally expressed miRNA is strongly limiting, but increasing studies will, hopefully, soon lead to the identification of a panel of miRNAs that can be safely used as reference controls.

Nonetheless, beyond these technical limitations, the most exciting but challenging application will be to utilize EVs and their cargo as a clinical tool to diagnose and monitor disease. Thus, EV-derived miRNAs could provide a novel therapeutic tool in RCC clinical practice, where the lack of adequate biomarkers makes functional investigations urgently needed.

## Figures and Tables

**Figure 1 biology-12-00877-f001:**
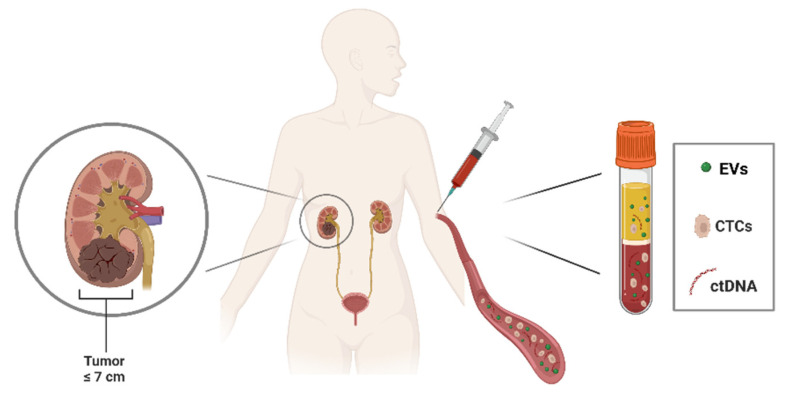
Peripheral blood collection in RCC patients. Liquid biopsy, in particular peripheral blood, can be a source of cancer material: circulating tumor cells (CTCs), circulating tumor DNA (ctDNA), and extracellular vesicles (EVs), released by tumors into the bloodstream. Figure was generated with BioRender.

**Figure 2 biology-12-00877-f002:**
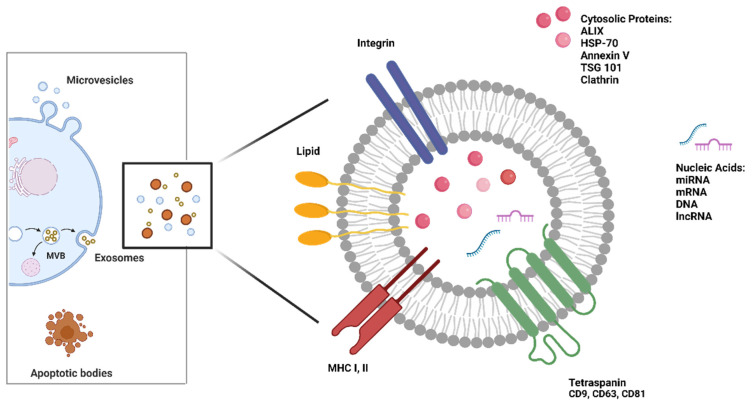
Biogenesis and structure of extracellular vesicles (EVs). EVs include three different sub-populations: exosomes are intraluminal vesicles released after fusion of a multivesicular body (MVB) with the cell membrane through exocytosis; microvesicles are formed by the outward shedding of the cell membrane into extracellular space; and apoptotic bodies are generated when cells undergo apoptosis. EV subtypes vary in size and composition and carry different cargoes such as nucleic acids and cytosolic proteins. They are also characterized by membrane proteins such as tetraspanins that identify the exosomal population. Figure was generated with BioRender.

**Figure 3 biology-12-00877-f003:**
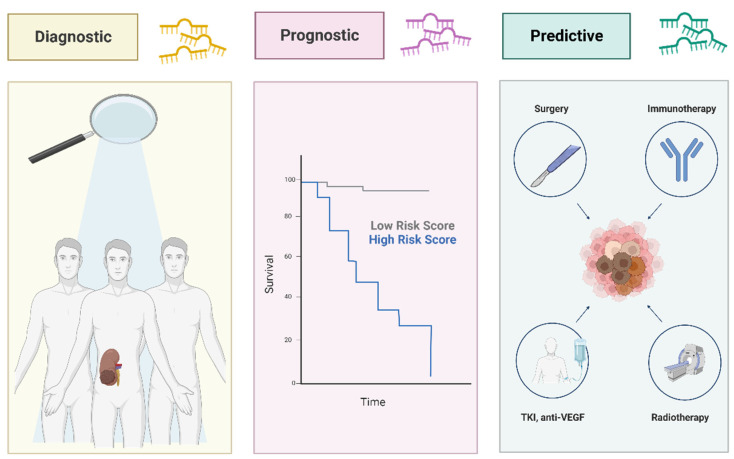
Circulating microRNAs (miRNAs) are potential biomarkers for RCC. miRNAs are emerging as promising biomarkers in RCC due to their potential triple role in clinical setting: several miRNAs are differentially expressed in tumors when compared with their normal counterpart, showing a valid diagnostic value. Experimental evidence has also revealed the potential prognostic value of several miRNAs for appraisal of survival of RCC patients; their promising role as predictors of response to therapy may acquire key importance in RCC, providing clinicians with crucial information to determine the best treatment plan. Figure was generated with BioRender.

## Data Availability

Not applicable.

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
