# Peer review of "Chasing the Role of miRNAs in RCC: From Free-Circulating to Extracellular-Vesicle-Derived Biomarkers"

_biology, 2023, doi:10.3390/biology12060877_

Round 1
Reviewer 1 Report
Dear Authors,
The manuscript by Ilena M et al. entitled: Chasing Circulating Biomarkers in Renal Cell Carcinoma: the Case of extracellular Vesicles and their miRNA Cargos, presenting the latest update about the circulating blood derived miRNAs and their future potential use as a biomarkers for early diagnosis of RCC, prognosis and their promising role as predictor of response to therapy. This is relevant not only in cancer biology and development understanding but also in clinical setting. The review is clear and presented in a well-structured manner.
However, tin the manuscript the reader has the impression that the authors are interested mainly in the role of miRNA in the RCC not in the EVs and their miRNA cargo. The authors need to emphasize on the role of EVs or at least clarify this point in their manuscript.
The authors can add a graphical abstract for better visibility and understanding of the mechanisms.
Authors can add one figure or edit the existing figures 1 or 2 where they can show how the circulating EVs-derived from RCC patients are isolated and characterized if possible (adding some markers,…) ?
Please check the abbreviations and add the corresponding name at least one time when is first cited in the manuscript’s text. E.g. KPS line 98, ICI line 115, VEGF-IO line 108.
Authors may clarify how Carbonic anhydrase enriched in exosomes stimulates angiogenesis.
They should also explain the role of azurocidin protein 1. Line 212-216.
In figure 3 MiRNA can be changed to miRNA. And the spelling can be checked in the whole text.
Table 1 (most of the research articles are focusing on the miRNAs role (only few published research about the EVs role).
Sincerely yours,
English is very clear can benefit for minor editing.
Reviewer 2 Report
11. The paper content doesn’t match the title of the article. The vast majority of the article is devoted to general information on renal cancer, extracellular vesicles, their biogenesis and role in RCC which is well-known, and only one part of the article is about role of miRNA.
22. The title of the article is «Chasing Circulating Biomarkers in Renal Cell Carcinoma: the Case of Extracellular Vesicles and their miRNA Cargos», but referring Table 1 the authors declare that the most of the studies analyzed the miRNAs expression within serum and plasma, without focusing on EV-derived miRNAs. In fact, there are a lot of investigations devoted to EV-derived miRNA and if you offer the article titled « Chasing Circulating Biomarkers in Renal Cell Carcinoma: the Case of Extracellular Vesicles and their miRNA Cargos» it is better to pay attention on EV-derived miRNA studies.
33. Line 373 - Several studies analyzed plasma and serum-derived miRNAs, without focusing on the vesicle compartment – doesn’t have strong basis, there are a lot of such studies published.
44. The conclusion repeats information from the main text and contain not enough authors’ own thoughts.
Round 2
Reviewer 2 Report
Authors did revisions